# NOT ALL MEMORIES ARE CREATED EQUAL: LEARNING TO EXPIRE

## ABSTRACT

Attention mechanisms have shown promising results in sequence modeling tasks that require long-term memory. Recent work has investigated mechanisms to reduce the computational cost of preserving and storing the memories (Rae et al., 2020). However, not all content in the past is equally important to remember. We propose *Expire-Span*, a method that learns to retain the most important information and *expire* the irrelevant information. This enables Transformers to scale to attend to tens of thousands of previous timesteps efficiently, as not all hidden states from previous timesteps are preserved. We demonstrate that Expire-Span can help models identify and retain critical information and show it can achieve state of the art results on long-context language modeling, reinforcement learning, and algorithmic tasks. Finally, we show that Expire-Span can scale to memories that are tens of thousands in size, which is helpful on incredibly long context tasks such as character-level PG-19 and a frame-by-frame moving objects task.

## 1 INTRODUCTION

Transformer architectures (Vaswani et al., 2017) have demonstrated strong performance across a variety of tasks (Devlin et al., 2019; Roller et al., 2020; Brown et al., 2020), including those that require learning long term relationships (Zhang et al., 2018; Fan et al., 2019a; Izacard & Grave, 2020). Recent work has focused on scaling attention mechanisms efficiently to longer memory sizes, enabling large improvements on long context tasks (Dai et al., 2019; Sukhbaatar et al., 2019a). However, a critical component of human memory is not just the ability to remember, but also *forgetting* irrelevant information to focus on the salient, relevant bits. Most studies of long-term memory in humans indicate that not everything is remembered (Murre & Dros, 2015) — instead, only vivid, remarkable memories are retained from the far past (Wixted, 2004).

Standard Transformer architectures lack the ability to search over extremely large memories, as the self-attention mechanism is computationally intensive and the storage cost of preserving the large memory grows quickly. Recent work (Child et al., 2019; Rae et al., 2020) has proposed learning how to extend to greater context through sparse mechanisms or through compression, to more compactly represent the past. However, there exists a fundamental problem with large memories beyond strict computational concerns: as the amount of information stored increases, deciding which information is relevant becomes more challenging. Other work (Lample et al., 2019) approaches this by considering how to efficiently search large memories. We will focus on learning what to forget, and thereby reducing the computational burden of the model easing the challenges of the search problem.

We propose EXPIRE-SPAN, a straightforward extension to attention mechanisms, that learns when to *expire* unneeded memories. By expiring memories that are no longer useful, EXPIRE-SPAN enables scaling to memories tens of thousands of timesteps long. This learnable mechanism allows the model to adjust the span size as needed, selecting which information is critical to retain and forgetting the rest. More concretely, we augment the self-attention with a simple predictor that outputs an expiration value for each hidden state that determines how long a memory should be retained and accessible to the model. After the EXPIRE-SPAN runs out, the memory will be forgotten, but in a gradually differentiable way to retain end-to-end training with backpropagation. This process is done independently for each layer, allowing different layers to specialize in different time-scales.

We demonstrate that EXPIRE-SPAN can distinguish between critical and irrelevant information on several illustrative tasks in natural language processing and reinforcement learning. We then show

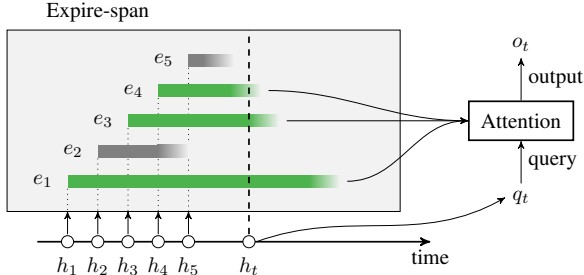

Figure 1: EXPIRE-SPAN. For every memory $h_i$, we compute an EXPIRE-SPAN $e_i$ that determines how long it should stay in memory. Here, memories $h_2$, $h_5$ are already expired at time $t$, so the query $q_t$ can only access $\{h_1, h_3, h_4\}$ in self-attention.

on long-context language modeling benchmarks and a frame-by-frame colliding objects task that EXPIRE-SPAN can scale to memories in the tens of thousands — by expiring irrelevant information, capacity is freed to have even larger memory. Finally, we analyze the information retained and expired by EXPIRE-SPAN models, to understand the importance of long context memory.

## 2 RELATED WORK

Memory is crucial for many tasks and has been studied in recurrent networks (Elman, 1990; Hochreiter & Schmidhuber, 1997; Mikolov et al., 2010) for a long time. The development of memory augmented networks (Graves et al., 2014; Sukhbaatar et al., 2015b) made it possible to store large quantities of information and selectively access them using attention (Bahdanau et al., 2015). The Transformer (Vaswani et al., 2017) took full advantage of this approach. Processing long sequences with Transformers is an active area with applications in language understanding (Brown et al., 2020), reinforcement learning (Parisotto et al., 2019), video processing (Wu et al., 2019), and protein folding (Rives et al., 2019). However, extending the memory span is computationally expensive due to the quadratic time and space complexity of self-attention.

Various work has focused on reducing this complexity and increasing memory capacity. Dynamic attention spans, such as Adaptive-Span (Sukhbaatar et al., 2019a) and Adaptively Sparse Transformer (Correia et al., 2019), focus on learning which heads can have shorter spans of attention, but can only extend to spans of a few thousand. Other work sparsifies attention by computing fewer tokens (Fan et al., 2019c), often by using fixed attention masks (Child et al., 2019) or sliding windows and dilation (Beltagy et al., 2020). The BP Transformer (Ye et al., 2019) structures tokens as a binary tree, so some tokens have coarse attention. These works focus on learning what to attend to, but searching larger and larger memories is very difficult. In contrast, we focus on learning to expire what is irrelevant. Compressive Transformer (Rae et al., 2020) reduces the number of memories by replacing every few memories with a single compressed one. A disadvantage of this is that all memories have the same compression ratio, so relevant memories are equally compressed.

Another line of work investigates linear-time attention mechanisms. Wu et al. (2018) replace self-attention with convolutions that run in linear time, but the scalability to long context tasks remains limited. Wang et al. (2020) propose linear time attention by decomposing attention into multiple smaller attentions, that recombine to form a low-rank factorization of the original attention. Those methods, however, focus on making attention more efficient without actually reducing the number of memories. Further, as our goal is to reduce the number of memories that feed to self-attention by learning to expire, EXPIRE-SPAN can be easily combined with these efficiency improvements. For a review of further recent Transformer variants, see Tay et al. (2020).

## 3 BACKGROUND

Transformer architectures have been widely used as decoder-only auto-regressive models for sequential tasks. Each Transformer decoder is made of a stack of identical layers, composed of a multi-head self-attention sublayer followed by a feedforward sublayer. The output of each timestep

is the hidden state $h_t^l$ at layer $l$, which is then projected to key $k$, value $v$, and query $q$ vectors:

$$q_t^l = W_q^l h_t^l, \quad k_t^l = W_k^l h_t^l, \quad v_t^l = W_v^l h_t^l.$$

where $W$ represents the weight. Going forward, we focus on a single layer and omit the layer index $l$ for brevity. Information from previous timesteps is accessed through attention $a$ to create output $o$:

$$a_{t,i} = \text{Softmax}_{i \in C_t} \left( q_t^\top k_i \right), \quad o_t = W_o \sum_{i \in C_t} a_{t,i} v_i.$$

The set $C_t$ indicates which memories can be accessed at time $t$, which is the focus on this work. The space and time complexity of self-attention is linearly correlated to $|C_t|$, making it an important metric of efficiency. For the rest of the paper, we will refer to $|C_t|$ as the *memory size*.

Including all previous timesteps in self-attention by setting $C_t = \{1, \ldots, t-1\}$ results in a quadratic complexity $O(T^2)$ to compute the full attention over a sequence of length $T$. *Fixed-spans* (Dai et al., 2019) take a more scalable approach such that $C_t = \{t - L, \ldots, t - 1\}$ so the attention is restricted to previous $L$ steps. The total complexity in this case is $O(TL)$, where $L$ is the attention span.

*Adaptive-Span* (Sukhbaatar et al., 2019a) further improves upon this by learning an optimal span $L$ per attention head from data, which results in small $L$ values for many heads. *Compression* approaches (Rae et al., 2020) reduce memory size by compressing multiple timesteps into a single memory, with complexity $O(TL/c)$, where $c$ is the compression rate. However, in all these approaches, all memories are treated equally without regards to their importance to the task. In this work, we focus on distinguishing between relevant and irrelevant memories by learning to expire unneeded information — by expiring, the remaining attention on relevant information can scale beyond existing long context memory approaches.

## 4 EXPIRE-SPAN

We detail the EXPIRE-SPAN mechanism and how to integrate it into Transformer architectures to focus attention on relevant information and expire the rest. We describe how to scale EXPIRE-SPAN and practically train with drastically longer memory spans.

### 4.1 METHOD

EXPIRE-SPAN, depicted in Figure 1, allows models to selectively forget memories that are no longer relevant. We describe it in the context of a single Transformer layer and omit the layer index $l$ for brevity. Our goal is to reduce the size of $C_t$ defined in Section 3 for more efficiency without performance degradation. For each memory $h_i$, we will compute a scalar EXPIRE-SPAN $e_i \in [0, L]$:

$$e_i = L\sigma(w^\top h_i + b).$$

Here $w, b$ represent trainable parameters, $\sigma$ is the sigmoid function, and $L$ is the maximum span. This expire-span $e_i$ determines how long $h_i$ should be kept and included in $C_t$.

At time $t$, the remaining span of $h_i$ is $r_{t,i} = e_i - (t-i)$. When $r_{t,i}$ becomes negative, it indicates the memory $h_i$ is expired and can be removed from $C_t$. This can be implemented by updating attention weights with a binary masking function $m_{t,i}$:

$$a'_{t,i} = \frac{m_{t,i} a_{t,i}}{\sum_j m_{t,j} a_{t,j}}, \quad o_t = \sum_i a'_{t,i} v_i \quad \text{where} \quad m_{t,i} = \begin{cases} 1 & \text{if } r_{t,i} > 0 \\ 0 & \text{otherwise,} \end{cases}$$

However, with such a discrete masking, the expire-span $e_i$ will not receive any gradient for training. Therefore, we use a masking function that smoothly transitions from 0 to 1 as shown in Figure 2:

$$m_{t,i} = \max(0, \min(1, 1 + r_{t,i}/R)),$$

where $R$ is a hyper-parameter that determines the length of a ramp that is bounded between 0 to 1. This function has non-zero gradient for values in $[-R, 0]$ to train $e_i$, but also can take a value of 0, which is necessary for expiring memories. Thus $C_t = \{i \mid m_{t,i} > 0\}$. Since $m_{t,i}$ is a monotonically decreasing function of $t$, once a memory is expired, it can be permanently deleted.

Our goal is to reduce average memory size, which is directly related with the average EXPIRE-SPAN.

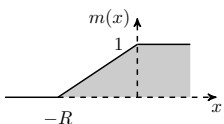

$$\frac{1}{T}\sum_t |C_t| = \frac{1}{T}\sum_t \sum_{i<t} \mathbf{1}_{m_{t,i}>0} = \frac{1}{T}\sum_i \left( R + \sum_{t>i} \mathbf{1}_{r_{t,i}>0} \right)$$

$$= \frac{1}{T}\sum_i \left( R + \sum_{t>i} \mathbf{1}_{e_i>t-i} \right) = R - 1 + \frac{1}{T}\sum_i \lfloor e_i \rfloor$$

Figure 2: Soft Mask Function

Thus, we add an auxiliary term to the loss function to penalize the L1-norm of EXPIRE-SPANS: $L_{\text{total}} = L_{\text{task}} + \alpha \sum_i e_i / T$, where $\alpha > 0$ is a hyperparameter. This loss term will decrease the span of memories that contribute less to the main task, resulting in a model with a small memory that focuses on the most relevant information. Note that $w$ and $b$ are the only new parameters, and are negligible in size compared to the total number of parameters in standard neural models.

### 4.2 ADDING EXPIRE-SPAN TO TRANSFORMERS

We describe how EXPIRE-SPAN can be utilized within Transformer self-attention layers to decrease the memory size and focus the memory on the most salient, relevant information, expiring the rest. We also discuss practical training concerns, such as efficiency and regularization.

**Modifications to Multi-Head Attention**    Self-attention consists of multiple heads that have different keys, values, and queries. However, they all share one underlying memory, so a memory cannot be removed if it is used by any of the heads. Thus, we compute an EXPIRE-SPAN at each layer that is shared amongst the heads.

**Block Parallel**    We use the caching mechanism (Dai et al., 2019), where a block of timesteps $B = [t, \ldots, t+b-1]$ is processed in parallel for efficiency — once a block is computed, its hidden states $[h_t, \ldots, h_{t+b-1}]$ are cached so that future blocks can attend to them. This means a memory can only be deleted if it is not used by any of the queries in $B$. Concretely, $h_i$ will be deleted when $m_{t,i} = 0$ where $t$ is the first token of $B$. However, this is not a concern for very long-term memories where $L \gg B$.

**Position Embedding**    Relative position embeddings (Shaw et al., 2018) make it possible to condition on the ordering of inputs by modifying the attention to $a_{t,i} = \text{Softmax}(q_t^\top k_i + q_t^\top p_{t-i})$. However, because this second term is computed for the whole block in parallel for efficiency, it can become expensive for a large $L$ even when the average memory size $|C_t|$ is small. Our solution is to remove position embeddings from older memories $i < t - b$ (where $b$ is the block size), which empirically does not affect performance. The computational complexity of the position embeddings is then $O(b)$, thus allowing us to increase the maximum span $L$.

**Loss Computation**    The L1-norm loss for EXPIRE-SPAN must be computed for every memory $h_i$. A simple way is to compute it on tokens within the current block $B$. This empirically results in poor performance — a possible explanation is that the time between positive and negative gradients on $e_i$ may become too distant. Negative gradients that increase $e_i$ only come from the main loss $L_{\text{task}}$ through the masking function $m_{t,i}$, which has non-zero gradients only when memory $h_i$ is about to expire with $0 < m_{t,i} < 1$ for $t \in B$. For a large $L \gg B$, $h_i$ may have been computed many blocks before and since then the model weights would have changed. In contrast, the positive gradients that decrease $e_i$ are computed on the current block $i \in B$. To remove this discrepancy, we compute the auxiliary loss on $e_i$ at the same time as negative gradients when $0 < m_{t,i} < 1$ for $t \in B$.

**Regularization**    A potential challenge in exceptionally long memory is greater capacity to overfit. As EXPIRE-SPAN can scale to memories in the tens of thousands, it can overfit to learning specific span sizes on the training set that do not generalize. As a form of regularization, we propose to randomly shorten the memory during training. For each batch, we sample $l \sim \mathcal{U}(0, L)$ and set $a_{t,i} = 0$ for all $t - i > l$ only during training. This way, the model cannot assume the memory will always contain specific information, as the memory is randomly shortened. This can be seen as a form of structured dropout (Fan et al., 2019b) applied to the memory size.

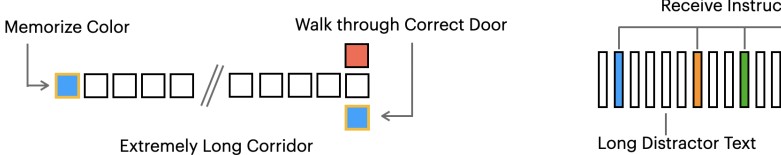

Figure 3: **Corridor Task (left)**- Agents must memorize the color of an object and walk through the door of the corresponding color at the end of a long corridor. **Instruction Task (right)**- A model must recognize instructions, memorize them, and execute when at the correct location.

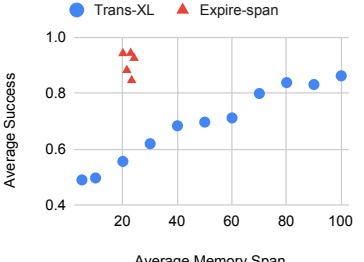 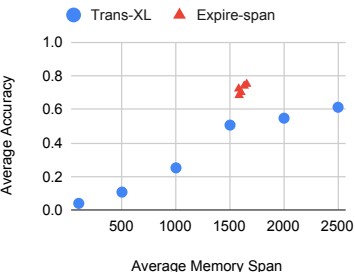

Figure 4: **Success on Corridor task as a function of the Memory Size (left)** — We trained 10 baseline models with different memory sizes, and five EXPIRE-SPAN models with different random seeds. Corridors are randomly sampled with length between $[3, 200]$. **Performance on Instruction task as a function of the Memory size (right)** — We trained 6 baseline models with different memory sizes, and five EXPIRE-SPAN models with different random seeds.

**Training with Small Initial Spans** EXPIRE-SPAN scales to long attention spans as it quickly learns to expire irrelevant content. However, at the beginning of training, the long span can use large quantities of GPU memory. To circumvent this, we initialize the bias term with a negative value. This prevents large memory usage at the beginning of training, after which the model quickly learns to expire and the memory usage is no longer problematic.

## 5 EXPERIMENTS AND RESULTS

We show that EXPIRE-SPAN can focus on salient information on various constructed tasks that necessitate expiration. Then, we highlight the scalability of EXPIRE-SPAN when operating on extremely large memories on different tasks. Additional experiments and details are in the Appendix.

### 5.1 BASELINES

We compare our method against several baselines from Section 3 that takes different approaches to limit the memory size. Transformer-XL (Dai et al., 2019) corresponds to the fixed-spans approach where simply the last $L$ memories are kept. Our Transformer-XL implementation also serves as a base model for all the other baselines to guarantee that the only difference among them is how memories are handled. The other baselines are Adaptive-Span (Sukhbaatar et al., 2019a) and Compressive Transformer (Rae et al., 2020). For Compressive Transformer, we implemented the mean-pooling version, which is shown to give a good performance despite its simplicity.

### 5.2 IMPORTANCE OF EXPIRATION: ILLUSTRATIVE TASKS

**Walking down a Corridor** To illustrate a case where proper expiration of unnecessary memories is critical, we begin with an RL gridworld task. In the Corridor task, depicted in Figure 3 (left), the agent is placed at one end of a very long corridor, next to an object that is either red or blue. The agent must walk down the corridor and go to the door that corresponds to the color of the object that it saw at the beginning to receive $+1$ reward. The requirement on the memory is low: the agent should remember the object color so it can walk through the correct door.

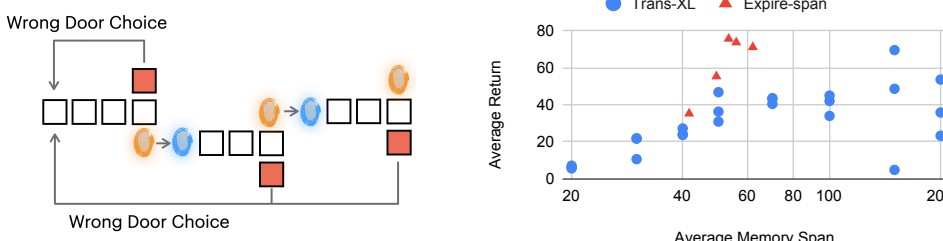

Figure 5: **Portal Task (left)**- An agent must trial-and-error to memorize the right sequence of doors. **Performance as a function of Memory Size on Portal Task (right)**- We train multiple models with different memory sizes and random seeds.

EXPIRE-SPAN models can take advantage of this fact and keep the memory size small regardless of the corridor length, which can vary between 3 and 200. This is confirmed in the results shown in Figure 4 (left) where the EXPIRE-SPAN models achieve high performance on this task with very small memories. Without the ability to forget, the Transformer-XL models require large memory for storing all navigation steps that grow with the corridor length.

**Multi-Room Portal Task**   Next, we analyze the performance of EXPIRE-SPAN on a reinforcement learning memorization task. Agents in a gridworld are tasked with navigating through multiple rooms separated by different doors, depicted in Figure 5 (left). Each room has two exit doors with different colors — one door portals to the adjacent room, while the other portals the agent back to the start. However, which door works in which room is randomized for each episode. Thus, the only way to visit more rooms is by trial-and-error, where agents need to remember the sequence of correct doors to successfully navigate to the end.

We show results in Figure 5 (right). Transformer-XL models need longer memory to perform better and visit more rooms — as each room requires many navigation steps, this requires more memory. However, those navigation steps are irrelevant because the agent only needs to memorize which doors are visited. Since EXPIRE-SPAN models can discard irrelevant memories and focus its memory on memorizing the exact sequence of doors, they achieve strong performance with much smaller memory compared to the Transformer-XL baseline.

**Receiving and Executing Instructions**   To illustrate a more difficult task where a model must learn to recognize relevant memories and expire the rest, we use a dialogue-based story generation task from the LIGHT (Urbanek et al., 2019) text world game environment. The model visits various locations and *receives* instructions of the form *can you tell the [butler] that the [town official] wants to see them?*. When the model is in a location where the *butler* is present, they must *execute* the instruction by generating *You tell the butler "town official wants to see you!"*. Between receiving and executing, thousands of words of distractor text exist. The model must learn what is relevant to retain and expire the distractors. Note multiple instructions can be in queue for execution.

We experiment with a dataset where the average distance between receiving and executing instructions is around 950 distractor sequences. Models are trained as language models, but evaluated only on their success in executing the instruction. Task details and model architecture are provided in the Appendix. We illustrate in Figure 4 (right) and Table 4 that EXPIRE-SPAN is much more successful at solving this task than Transformer-XL, Adaptive-Span, and Compressive Transformer as it can focus on the specific instruction lines. For Adaptive-Span and Compressive Transformer, we experiment with various memory sizes to tune the maximum span.

## 5.3   SCALABILITY OF EXPIRE-SPAN

We analyze the scalability of EXPIRE-SPAN. On a copy task, we train models with spans up to 128K timesteps. Then, we show the utility of EXPIRE-SPAN on two character-level language modeling tasks — enwiki8 and PG-19, and colliding objects task that is processed frame by frame.

**Extremely Long Copy**   To illustrate the scalability of EXPIRE-SPAN, we construct a copy task where the model sees a sequence of *A* very far in the past. The rest of the characters are *B*. The

| Model | Copy Acc (%) |
|---|---|
| Transformer-XL | 26.7 |
| EXPIRE-SPAN 16K | 29.4 |
| EXPIRE-SPAN 128K | 52.1 |

Table 1: **Copy Task.** We report copy accuracy on the test set.

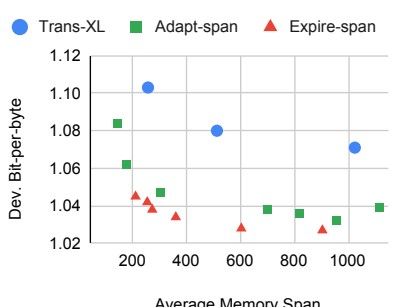

Figure 6: **Performance on character-level PG-19.** We report bit-per-byte on test.

| Model | Params | Test |
|---|---|---|
| Trans-XL 24L (Dai et al., 2019) | 277M | 0.99 |
| Sparse Trans. (Child et al., 2019) | 95M | 0.99 |
| Adapt-Span 24L (Sukhbaatar et al., 2019a) | 209M | 0.98 |
| All-Attention (Sukhbaatar et al., 2019b) | 114M | 0.98 |
| Compressive Trans. (Rae et al., 2020) | 277M | 0.97 |
| Routing Trans. (Roy et al., 2020) | - | 0.99 |
| Feedback Trans. (Fan et al., 2020) | 77M | 0.96 |
| Trans-XL 12L (Dai et al., 2019) | 41M | 1.06 |
| Adapt-Span 12L (Sukhbaatar et al., 2019a) | 39M | 1.02 |
| Our Trans-XL 12L baseline | 38M | 1.06 |
| EXPIRE-SPAN 12L | 38M | 0.994 |

Table 2: **Enwiki8 Results.** We report bit-per-byte (bpb) on test and the number of parameters.

Figure 7: **Performance as a function of Memory Size on enwiki8.**

model must copy the correct quantity of $A$. We design the task such that long span (up to 128K) can be required, as the $A$ tokens are very far into the past. In Table 1, we show that only by scaling the attention span to 128K is it possible to achieve improved performance. We compare to a Transformer-XL baseline with 2048 attention limit and a EXPIRE-SPAN model with smaller span.

**enwiki8** We test on the enwiki8 character-level language modeling benchmark (Mahoney, 2011). We set the maximum span $L$ for EXPIRE-SPAN to 16K. We compare EXPIRE-SPAN to existing work in Table 2. EXPIRE-SPAN outperforms similarly sized baselines by a large margin, and matches the performances of much larger models. This indicates that models can learn to expire relevant information and encode long context effectively, even on competitive language modeling benchmarks.

Next, we compare the performance of EXPIRE-SPAN with Adaptive-Span and Transformer-XL baselines when we vary the average span size (see Figure 7). Models with EXPIRE-SPAN achieve stronger results — when comparing at any given memory size, EXPIRE-SPAN has better performance than both baselines. Further, the performance of models with EXPIRE-SPAN does not vary widely even if the memory size is drastically reduced[1].

Finally, we compare the performance of EXPIRE-SPAN with the Compressive Transformer baseline. We vary the memory size of Compressive Transformer and compress with rate 4. We find that EXPIRE-SPAN models achieve much better performance, as shown in Table 4 with less GPU memory and faster training time per batch.

**Character Level PG-19** We use the PG-19 (Rae et al., 2020) language modeling benchmark and convert it to model characters. We use all characters in the training set, creating a vocabulary size of 3506 characters. We compare the performance of Transformer-XL with an attention over the last 1K and 2K timesteps with an EXPIRE-SPAN model with maximum spans at 8K, 16K, and 32K. As shown in Figure 6, EXPIRE-SPAN models train stably with very long span, and EXPIRE-SPAN is substantially better than the Transformer baseline. The performance improves with larger spans,

---

[1] The results in Figure 7 are not finetuned with a smaller learning rate, thus have slightly worse performances than Table 2

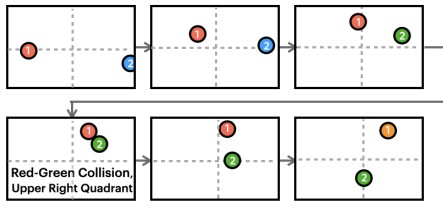

Figure 8: **Object Collision**: task is to remember the location of specified colored collisions.

| Model | Maximum Span | Test Error (%) |
|---|---|---|
| Trans-XL | 1k | 73.3 |
| Compressive | 8k | 63.8 |
| Adapt-Span | 16k | 59.8 |
| Expire-Span | 16k | 52.2 |
| Expire-Span | 32k | 36.7 |
| Expire-Span | 64k | **26.7** |

Table 3: **Colliding Objects Results**. We report error on the test set.

| | Model | Performance | GPU Memory | Time/Batch |
|---|---|---|---|---|
| Instruction Task | Compressive Transformer | 71% Acc | 9.8 GB | 210 ms |
| | Adaptive-Span | 64% Acc | 14.2 GB | 240 ms |
| | EXPIRE-SPAN | **74% Acc** | **8.4 GB** | **90 ms** |
| Collision Task | Compressive Transformer | 63.8% Error | **11.6 GB** | 327 ms |
| | Adaptive-Span | 59.8% Error | 17.2 GB | 365 ms |
| | EXPIRE-SPAN | **52.2% Error** | 11.9 GB | **130 ms** |
| Enwiki8 | Compressive Transformer | 1.015 bpb | 21.3 GB | 805 ms |
| | Adaptive-Span | 1.036 bpb | 20.3 GB | 483 ms |
| | EXPIRE-SPAN | **1.034 bpb** | **14.6 GB** | **408 ms** |
| PG19 | Adaptive-Span | **1.12 bpc** | 13.5 GB | **515 ms** |
| | EXPIRE-SPAN | **1.12 bpc** | **12.9 GB** | 585 ms |

Table 4: **Efficiency of EXPIRE-SPAN**. We report peak GPU memory usage and per-batch training time, fixing the batch size. We evaluate the mean pooling version of the Compressive Transformer.

though extending to 32K does not increase performance beyond 16K. When comparing to Adaptive Span in Table 4, Adaptive-Span has the same performance but requires slightly more memory while running bit faster.

**Colliding Objects**   Another setting where learning which long context may be important is in video understanding. Despite video data being memory intensive, salient events might be localized in space and time. We test our model on a task where two objects move around and collide, and the goal is to reason about the location of specified-color collisions. Objects have a color that can randomly change. We divide a grid into four quadrants and the model is asked to recall the quadrants of the last collision of a specified-color pair. Because the collisions are rare, and collisions of specific colors are even rarer, the model must process a large quantity of frames.

We display the task in Figure 8 and results in Table 3. The task requires many frames, so long context is very beneficial — we see that as the EXPIRE-SPAN maximal span increases, performance steadily rises. We train the 64K with the random drop regularization method described in Section 4.2. When extending to extremely long span, 64k in size, we find the strongest performance, which matches the size of the largest attention limit reported to date (Kitaev et al., 2019).

As a baseline, we train an Adaptive-Span model with maximum spans limited to 16k. While this model perform similar to the 16k EXPIRE-SPAN model, it uses 44% more memory (17.2gb compared to 11.9gb) and takes $2.8\times$ longer to train on a batch (365ms compared to 130ms). A 32k Adaptive-Span model run out of memory in the same setting where we trained our 64k EXPIRE-SPAN model, which shows the efficiency of our approach.

We also included a Compressive Transformer baseline with a maximum span of 8.5k (512 normal memories and 2k memories compressed by $4\times$). This model performs worse than our model, and increasing its span does not help. It uses the same GPU memory as our 16k model, but runs $2.5\times$ slower, despite having a smaller span.

## 5.4   EFFICIENCY OF EXPIRE-SPAN

We describe the efficiency of EXPIRE-SPAN compared to various baselines, including Adaptive-Span, Compressive Transformer, and Transformer-XL. We quantify with two metrics: **(1)** Peak

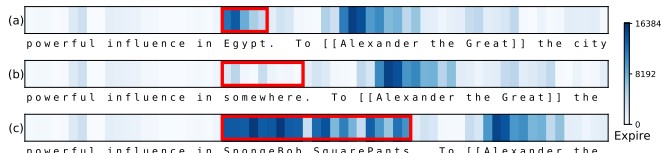 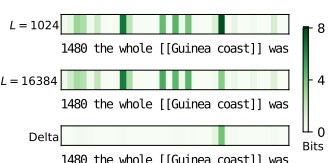

Figure 9: **Expiration in EXPIRE-SPAN on enwiki8**. In (a), one layer of the model has two focus areas, "Egypt" and "Alexander". In (b), we replace "Egypt" with "somewhere" and the focus disappears. In (c), we insert "SpongeBob SquarePants" and the model attends to the full entity.

Figure 10: **Prediction Accuracy Needs Memory.** As the memory size is artificially decreased at inference to 1024, the prediction is less accurate (smaller is better).

GPU memory usage and (2) training time per batch (comparing fixed batch size). Results are shown in Table 4 and indicate that across all tasks, EXPIRE-SPAN uses substantially less GPU memory and processes each batch more quickly than all baselines, while achieving better performance in most of the tasks. Transformer-XL as a baseline cannot adapt to the data at all, so it becomes slow and inefficient quite quickly. Adaptive-Span can adapt to the data and adjust its memory size, but this memory size is fixed after training and does not have the dynamic adjustment of Expire-Span (where memory depends on context). Finally, Compressive Transformer does compress the past memories, but it compresses them all the same amount. In contrast, EXPIRE-SPAN can forget irrelevant content, which both improves performance by focusing on salient information, but also reduces the load on the GPU and allows for faster processing per batch. Overall, across all tasks, we find that EXPIRE-SPAN achieves strong performance with GPU memory savings and low time per batch.

## 6 ANALYSIS AND DISCUSSION

We analyze the information retained and expired by EXPIRE-SPAN models. See the appendix for more analysis and ablations on the importance of regularization.

**Retaining Salient Information**   We analyze what is retained by an EXPIRE-SPAN model on enwiki8. In Figure 9 (a), we show that the model retains information about named entities *Egypt* and *Alexander the Great* by attending to them (darker color). Next, we analyze how attention changes when we artificially edit the past text. In Figure 9 (b), we replace *Egypt* with *somewhere*, and this generic word is expired. In Figure 9 (c), we edit *Egypt* to *Spongebob SquarePants*, which is a rare named entity, and the model attends strongly to this rather than expiring. In addition to named entities, EXPIRE-SPAN also focuses on spaces, newlines, and section titles, all of which can retain information about words, sentences or sections. These can vary between different layers, indicating that EXPIRE-SPAN models use the memory at each layer differently. See the Appendix for details.

**Importance of Long Term Memory**   Lastly, we analyze which predictions benefit the most from memory capacity. We take an EXPIRE-SPAN model trained on enwiki8 and artificially limit the span size at inference. We compare which predictions decreased in accuracy. In Figure 10, we limit the span to 1024 tokens, and see that models have a higher loss when predicting the named entity *Guinea coast* compared to the 16k span. *Guinea coast* was mentioned 3584 tokens earlier, which indicates that long attention is often necessary to predict words mentioned in far away context. In general, we found that rare tokens and structural information about documents, such as section headings or document titles, required longer attention span.

## 7 CONCLUSION

We present EXPIRE-SPAN, an operation that can be added to any attention mechanism to enable models to learn what to forget. By expiring irrelevant information, models can scale attention to tens of thousands of past memories. We highlight the strong performance of EXPIRE-SPAN in language modeling, reinforcement learning, object collision, and algorithmic tasks, and use it to attend to over tens of thousands of past memories. The scalability of EXPIRE-SPAN has strong potential for allowing models to be applied to more challenging, human-like tasks that would require expiration.

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

# A APPENDIX

## A.1 ADDITIONAL EXPERIMENTAL RESULTS

### A.1.1 WIKITEXT-103 LANGUAGE MODELING

The Wikitext-103 word-level language modeling benchmark (Merity et al., 2016) consists of a collection of Wikipedia articles and a fixed vocabulary size of 270K. We set the max attention span for EXPIRE-SPAN to 8K. We compare EXPIRE-SPAN to existing work in Table 5 and show that even fairly small models trained with EXPIRE-SPAN achieve competitive results. Next, we analyze the performance of EXPIRE-SPAN on Wikitext-103 as the memory size increases. We compare to a standard Transformer model in Figure 11 — even with far smaller memory, EXPIRE-SPAN performs much better.

### A.1.2 EXPIRE-SPAN ANALYSIS ON ENWIK8

In Figure 12, we analyze multiple layers of the model and show that different layers identify different types of information. Several layers retain summarizing information about sentences or sections by attending strongly to spaces, new lines, and section titles.

### A.1.3 IMPORTANCE OF STRUCTURED DROPOUT FOR REGULARIZATION

We analyze the importance of structured dropout to regularize the large memory capacity provided by EXPIRE-SPAN. In an experiment on enwiki8, Figure 13 shows that loss on a portion of validation data was incredibly large. This part corresponds to a 66K token long table. We hypothesize that the model likely never encountered such a table during training. During validation, this caused all non-table tokens to expire. Without regularizing the model memory size during training, the model can easily overfit.

| Model | Params | Test |
|---|---|---|
| DEQ-Trans. (Bai et al., 2019) | 110M | 23.3 |
| Trans-XL (Dai et al., 2019) | 257M | 18.3 |
| Feedback Trans. (Fan et al., 2020) | 77M | 18.3 |
| Trans.+LayerDrop (Fan et al., 2019b) | 423M | 17.7 |
| Compressive Trans. (Rae et al., 2020) | 277M | 17.1 |
| Routing Trans. (Roy et al., 2020) | - | 15.8 |
| EXPIRE-SPAN | 140M | 19.6 |

Table 5: **Wikitext-103 Results.** We report perplexity on test.

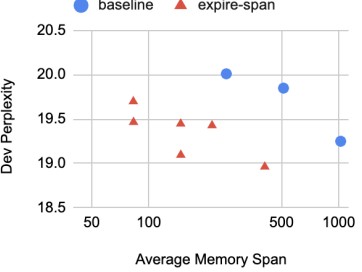

Figure 11: **Performance as a function of Memory Size on Wikitext-103**

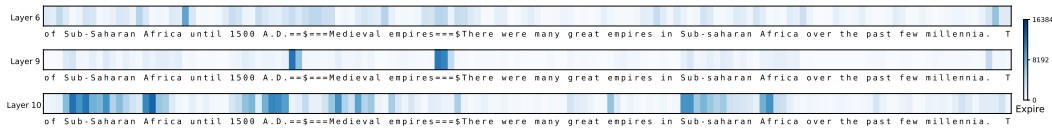

Figure 12: **Per-Layer Expiration in EXPIRE-SPAN on enwiki8**. We visualize the expire-spans of different layers: layer 6 gives long span to spaces, layer 9 favorites special tokens like newlines and section titles, and layer 10 pays attention to named entities.

### A.1.4 COLLIDING OBJECTS EASY

We experiment with an easier version of the Colliding Objects task where objects do not have colors. The model has to predict either the last collision, or a mapping of the last 3 collisions. In contrast to the harder task, there are no color switches and any collision prediction is valid.

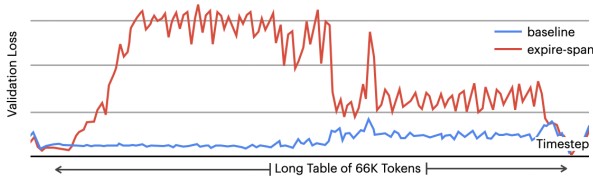

Figure 13: **Extreme Overfitting** on part of validation occurs without proper regularization.

| Model | Maximum Span | Test Error |
|---|---|---|
| Transformer | 1024 | 39.1 |
| Expire-Span | 1024 | 19.5 |
| Expire-Span | 2048 | 9.1 |
| Expire-Span | 4096 | 3.2 |

Table 6: **Colliding Objects Results**. We report test error.

## A.2 ADDITIONAL IMPLEMENTATION DETAILS

### A.2.1 LIGHT TASKS

**Instruction Following Task** We train 6-layer models with a hidden size of 512 and 8 attention heads. To train, we use the Adam optimizer with a learning rate of 7e-4 and 8000 warmup updates. We set the Expire-Span ramp to 64 and the Expire-Span loss to 2e-6.

### A.2.2 REINFORCEMENT LEARNING

We used MazeBase (Sukhbaatar et al., 2015a) to construct tasks in grid world. Agents can observe its surrounding $3 \times 3$ area and move in the four cardinal directions. Every objects and their properties are described by words such as "agent","block", "blue", etc. Thus, the input to the model is a binary tensor of size $3 \times 3 \times$ vocabulary-size.

We train 2-layer Transformers with 64 hidden units using actor-critic algorithm. We used a BPTT length of 100, and an entropy cost of 0.0005.

**Corridor Task** The corridor length is sampled from $\mathcal{U}(3, 200)$. All models are trained for 100M steps. We used RMSProp optimizer with a learning rate of 0.0001 and a batch size of 64. For the expire-span models, we set the maximum span $L$ to 200, the loss coefficient $\alpha$ to 5e-6 and the ramp length $R$ to 16.

**Multi-Room Portal** In this task, there are 50 rooms sequentially connected together. Each room is $5 \times 5$ in size, and have two doors with different colors. If agent go to the correct door, it will be teleported to the next room, but if it is the wrong door, the agent will be teleported back to the first room and have to start over. Which of the two doors is correct in each room is randomly decided and fixed throughout the episode. This information is not visible to the agent, thus can only be discovered by trial and error within each episode. The current room number is visible to the agent.

When the agent successfully transitions from the $k$-th room to the next, it receives a reward of $0.1k$. The episode ends if the agent makes two mistakes in the same room, reaches the last room, or when the number of steps reach 1000. A reward discount of 0.98 is used. All models are trained with Adam optimizer with a learning rate of 0.0005, and a batch size of 1024, with gradients are clipped at 0.1. We set $L = 100$, $R = 16$ and $\alpha =$1e-6 for the expire-span models.

### A.2.3 COLLISION TASK

At the start of the simulation, each particle samples a Gaussian Normal velocity and position uniform inside a $16 \times 16$ grid. At each time step the particles' position is updated by adding its velocity (unless it would go off the grid, in which case its velocity is re-sampled). There are 5 different colors, and a

particle can change its color randomly at each step with 0.05 probability. A collision happens when the two particles have the same rasterized locations, but it does not affect the movement.

Given a question specifying two colors, the task is to report in which of the four quadrants of the grid the last collision of the specified-colors occurred. To make the task easier to learn, 40% of the queries will have the matching colors as the last collision.

The model is given an input sequence of tokens that has 8 entries per timestep. The first 4 are the rounded and rasterized $(x, y)$ locations of the two particles, and next 2 are tokens representing the colors of the particles. The last 2 entries are "question" tokens that specify the colors of the collision. The output sequence has a token for each quadrant. We generate 50M steps for training, which equals to 400M entries.

**Easy Version:** The particles have no color in this version. There are two types of questions, in which the task is to report either

1. in which of the four quadrants of the grid the last collision occurred.
2. the label mapping of the last 3 collisions.

### A.2.4 LANGUAGE MODELING DETAILS

**enwiki8** We train 12-layer models with a hidden size of 512 and 8 attention heads. To train, we use Adam optimizer with a learning rate of 7e-4, a batch size of 512, a block size of 512 and 8000 warmup updates. All models are trained for 100k updates. The model in Table 2 is further fine-tuned for another 10k updates with a 10x smaller LR. The baseline models used for comparison are the same size model following the same training protocol.

**PG-19** Besides the attention span, all model parameters and training parameters were held constant. Each model had 12 layers, a hidden size of 512, a feed forward hidden size of 2048, 8 attention heads, and processed 512 characters at a time. We initialized the weights using a uniform distribution as described by Glorot & Bengio (2010), used dropout and attention dropout of 0.2, clipped the gradients at 0.3, warmed up the learning rate linearly for 8000 steps, and used cosine annealing to decay the learning rate after warmup (Loshchilov & Hutter, 2016). For the EXPIRE-SPAN models, we used a ramp of 128 and an expiration loss coefficient of 1e-6.

**Wikitext-103** All models have 8 layers and 1024 hidden units (4096 in ReLU layers). In addition to the dropout of 0.3 applied to attention and ReLU activation, outputs from the embedding layer and the last layer had a dropout of 0.2. We used the adaptive input Baevski & Auli (2019) and the adaptive softmax Grave et al. (2017) for reducing the number of parameters within word embeddings. The models are trained for 300k updates with a block size of 256, and gradients are clipped at 0.1. The other hyperparameters are the same as the enwik8 experiments.

