# OpenReview forum: "Not All Memories are Created Equal: Learning to Expire"
_ICLR.cc/2021/Conference — Reject_

### Official Review · AnonReviewer1 · 2020-10-28
**Overcoming the long-term memory bottleneck of transformers**

**Rating:** 5
**Confidence:** 4

**Review:**

**Summary**
The paper proposes a method for overcoming the long-term memory bottleneck of transformers. The idea is to assign a value (expire-span) to each formed memory, which indicates how long the memory should be stored and be available for the transformer to access it. The authors demonstrate the performance of their approach on a set of
synthetic and character-level language modeling benchmarks.

**Significance**
While the idea seems to be quite interesting and the presentation of the paper is clear and sound, I have the following concerns:

- As the expire-span does not seem to be updated, the model must know how long to keep the memory when the memory is formed. Couldn't this potential cause issues when information arriving in the future would influence
the span of how long the memory should be kept?
- From the author's descriptions, the method appears relatively brittle to hyperparameter choice. In particular, the method requires some sophisticated form of regularization for the performed benchmarks. Thus, raising my concerns about the stability and scalability of the approach.

I would appreciate it if the authors could elaborate on my concerns.

- The paper misses important related work in this domain. The paper Gers et al. "Learning to forget continual prediction with LSTM" already proposes a mechanism to remove memories that are not needed anymore. Moreover, the proposed approach is adaptive as for each token, the network decides if it should clear some of its memory.

---

> ### Author Response · Authors · 2020-11-23
> **review response**
>
> Thanks for your review. We’ve responded to your points in detail below:
>
> **re: How does the model understand how long to keep the memory?**
>
> The Expire-Span values are not directly affected by future events, but they are being updated during training. If a future event that requires a specific memory occurs within its ramp steps (fixed R steps after expiring), it still can access that memory. The ramp suppresses the memory in a differentiable way, so the future event can cause the expire-span to grow during training.
>
> **re: model is brittle to regularization hyperparameters**
>
> We introduce only one new hyperparameter, which is the loss coefficient for penalizing long spans. This parameter is actually useful in controlling the trade-off between computation and performance. A lower loss coefficient means longer spans, which are good for performance but will use more compute. If compute is limited, then a higher loss should be used to reduce spans, which might have a negative effect on performance. Other Transformers have such a parameter too: TransformerXL has a fixed “attention length” parameter that does the same thing and needs to be tuned.
>
> Regarding regularization --- in general, neural networks require regularization to train them stably. We identified this regularization issue and propose a straightforward way to fix it. In general, we feel this is not very different from e.g. a specialized learning rate schedule. This capability to forget memories to extend to very long context is a novel contribution, and understanding how to properly regularize is part of our work in showing how to train such a model. These ideas are generally useful for others interested in this direction.
>
> **re: missing important work from Gers et al**
>
> Thanks for the reference, we will include in the paper. However, our approach targets attention on external memory because of their computation complexity. Forgetting reduces the number vectors to attend over, so it reduces computation and memory footprint. In LSTMs, however, memory is always a single vector, and forgetting means resetting that vector. So forgetting in a LSTM does not make it run faster, or reduce its memory footprint.

---

### Official Review · AnonReviewer2 · 2020-10-28
**Impactful Idea with Clear Exposition, but Experiments Fall Short**

**Rating:** 6
**Confidence:** 3

**Review:**

Modified score: thank you authors for your thorough response. Given the new information and baselines, I think this is a promising paper that passes the acceptance threshold.

Overall Quality: The authors present a method to improve the efficiency of transformer models when computing attention over previous time steps. Although this presents a neat idea that has the potential to improve an increasingly important model architecture, the experiments fall short of matching the claim that this method provides enables more efficient attention computation over memories _in practice_. Specifically, their baselines do not include relevant transformer modifications aimed at efficiency and they provide no detailed analysis on the memory size in practice. If the authors included more thorough experiments, this would be a strong paper. In their absence, it is marginally below the acceptance threshold.

Clarity: The abstract, introduction, background, and methods section were detailed yet easy to follow. The comparison of time complexity of prior work in the background section was particularly helpful. However, this precision did not carry over into the experimental section, which lacked thorough experimentation (detailed under weaknesses below) and figures 3-5 were out of order relative to the prose (the latter point is minor and does not affect my rating).

Significance: The potential impact is very high, especially as applications for transformers grow. If the authors could address the weaknesses outlined below, this could be an enormously helpful augmentation to the transformer architecture.

Strengths:
- The authors focus on an important problem for a very relevant architecture.
- The writing is clear and enjoyable. Section 3 in particular is a very friendly introduction to transformer time complexity.
- Evaluations performed over a variety of applications, spanning simple/toy to more realistic tasks.

Weaknesses:
- Corridor, instruction, portal, copy, pg-19, and colliding objects tasks only show comparisons for standard transformer models, as opposed to (at least one or two) comparable efficiency-optimized models. Giving the authors the benefit of the doubt, the first few experiments may serve more as proofs of concepts, where direct comparison with prior work is not as relevant or useful. But this leaves only one task in the paper with comparison to prior work on improving transformer efficiency: en-wiki-8. On en-wiki-8, the authors compare with just 1 modification and the improvement seems rather small. Small margins of improvement alone are not enough to reject a paper, but, given that this is the only result with a head to head comparison of efficiency optimized transformers, it makes it difficult for the community to discern the contribution of the work. Furthermore, on pg-19, copy task, and object collision, the authors do not provide the memory size/average memory size/effective memory size. This makes it difficult to understand if performance gains correspond with performance improvements, which is the methods stated purpose.
- Intuitively, an inductive bias to expire memories would make a learned model more brittle when transferring to new tasks. E.g., in the instruction task a new form of instruction may become relevant in a test task that was never relevant in training tasks Why is this a reasonable trade-off to make?

Question:
- What value is shown in table 2? The caption says bit-per-byte, but the numbers are inconsistent with figure 7.
- in figure 11, how is memory computed?

---

> ### Author Response · Authors · 2020-11-23
> **review response**
>
> Thanks for your review! We are happy to see that you find the idea impactful. Below, we’ve responded to each of your questions.
>
> **re: many tasks only have comparisons to standard transformers and there is no efficiency analysis**
>
> Thanks! Based on your feedback, we have added several baselines for multiple tasks (including PG-19, per your suggestion to add results for a real task), and have completed a detailed efficiency analysis for all baselines and Expire-Span. We quantify both time to process a batch and peak GPU memory usage to capture efficiency. These results are in the general response and added to our paper.
>
> **re: does expiration make the learned model more brittle during transfer learning?**
>
> We haven’t tested our method in a transfer learning setting. But, if a model can be trained on the new task, then expire-span can learn to change and adapt to new requirements, just like the rest of the network. If there is no training on the new task, then a model without expire-span is brittle too because self-attention can assign 0 probability to important memories that are not used during training.
>
> **re: what value is shown in Table 2? It’s inconsistent with Figure 7**
>
> They are both bit-per-byte, but the Table 2 number is further optimized with a smaller learning. This finetuning of a final model (or using decaying learning rate schedule) is a common practice employed by the other baselines.
>
> **re: in figure 11, how is the memory computed?**
>
> It is computed in this way: for each query, we count key and value vectors it attends to (i.e., by excluding the expired memories), then average over all queries. This measure directly correlates to the number of FLOPS and the memory usage.

---

### Official Review · AnonReviewer4 · 2020-10-28
**A solid improvement of Transformer's attention mechanism, yet the baselines and empirical results are not strong**

**Rating:** 6
**Confidence:** 4

**Review:**

To help Transformer learn long sequence efficiently, the paper performs attention on selective timesteps that have high expire-span scores. For each timestep, the expire-span score is computed by mapping the corresponding hidden feature to a number, which is learnt during training. Soft masking is applied to make the learning differentiable. An additional loss is introduced to reduce the average span, making the attention sparse. The proposed attention is integrated into each layer of Transformer and tested on several synthetic tasks and two language modelling datasets, yielding promising results.

Pros:
- The proposed solution (computing expire score and minimizing the average span) is elegant and seems novel within Transformer context
- The properties and behaviours of the method are well illustrated with detailed analysis and visualization
- Diverse experiments are conducted

Cons:
- Insufficient comparison with other Transformer-based baselines
- The results on real data are weak

Detail comments and questions

- Sec 4.1, the equation computing o_t should be a summation over i
- Before Transformer, sparse attention has been studied deeply in the literature. It may be beneficial to review some works (e.g., [1,2,3]) and try to integrate them into Transformer as additional baselines to make the experiment stronger.
- No experimental result demonstrates that the method can reduce computation complexity. Please consider including a comparison of running time or physical memory usages between your method and other Transformers
- It is unclear what are the baselines mentioned in the experiments. Are they vanilla Transformers? How did the authors control the memory size of the baseline as in Fig.3, 4 and 7?
- For some synthetic tasks, it is better to include stronger baselines [4,5] to show the advantage of the proposed method over other variants of Transformer
- In Table 2, the performance gap is significant. Is it possible to improve your performance with more parameters?

[1] Ke, Nan Rosemary, Anirudh Goyal ALIAS PARTH GOYAL, Olexa Bilaniuk, Jonathan Binas, Michael C. Mozer, Chris Pal, and Yoshua Bengio. "Sparse attentive backtracking: Temporal credit assignment through reminding." In Advances in neural information processing systems, pp. 7640-7651. 2018.

[2] Martins, Andre, and Ramon Astudillo. "From softmax to sparsemax: A sparse model of attention and multi-label classification." In International Conference on Machine Learning, pp. 1614-1623. 2016.

[3] Niculae, Vlad, and Mathieu Blondel. "A regularized framework for sparse and structured neural attention." In Advances in neural information processing systems, pp. 3338-3348. 2017.

[4] Gonc¸alo M Correia, Vlad Niculae, and Andre FT Martins. Adaptively sparse transformers. In Proceedings of the 2019 Conference on Empirical Methods in Natural Language Processing and the 9th International Joint Conference on Natural Language Processing (EMNLP-IJCNLP), pp. 2174–2184, 2019.

[5] Sainbayar Sukhbaatar, Edouard Grave, Piotr Bojanowski, and Armand Joulin. Adaptive attention ´ span in transformers. In Proceedings of the 57th Annual Meeting of the Association for Computational Linguistics, pp. 331–335, 2019a.

---

> ### Author Response · Authors · 2020-11-23
> **review response**
>
> Thanks for your review and all of the detailed questions!
>
> **re: insufficient comparison with baselines**
>
> Based on your feedback, we have added several baselines for multiple tasks and have completed a detailed efficiency analysis for all baselines and Expire-Span. We quantify both time to process a batch and peak GPU memory usage to capture efficiency. These results are in the general response and added to our paper.
>
> **re: equation in section 4.1 requires a summation over i**
>
> Thanks, fixed!
>
> **re: can you add sparse attention as a baseline?**
>
> Our goal is to improve the efficiency of long-term memory. In contrast, [1,2,3] focus on different aspects of sparse attention (improved gradient flow, more interpretable improved generalization). For example, [1] does top-k sparse attention, which still requires attention over all memories first. The same is true for sparsemax in [2] and [3], which is not more efficient than the full attention.  Please see our general response for some additional details.
>
> **re: efficiency analysis**
>
> In the general response, we have added exactly your suggestion, to compare running time and physical memory usage. Thanks for the constructive feedback!
>
> **re: is the Transformer baseline a vanilla Transformer? How do you control the memory size of this baseline?**
>
> Our vanilla Transformer baseline is actually based on TransformerXL (caching mechanism, relative position embedding) --- we have now clarified this in the paper. The memory size is referred to as “attention length” in TransformerXL, which is a hyperparameter that can be adjusted. It simply restricts how many previous tokens can be attended at time t.
>
> **re: for the synthetic tasks, please include stronger baselines**
>
> We have added baselines for many tasks, synthetic and not synthetic. We experimented with the Compressive Transformer and Adaptive-Span baselines to supplement our original Transformer-XL comparisons. Please see the general response, thanks!
>
> **re: performance gap in Table 2, can you make models larger?**
>
> Our goal is not to set a new SOTA, but rather propose an efficient model with a good performance. We added baselines similar to our model in size to Table 2, and our model outperforms them by a large margin. Training a large model is possible, but it requires more GPUs and careful tuning of regularization parameters, which takes a lot of resources. (note that the current SOTA, the Feedback Transformer, is actually a less efficient architecture that takes much longer to train)

---

### Author Response · Authors · 2020-11-23
**General Response**

We thank the reviewers. Expire-Span provides the general capability for models to learn what is important to remember and forget in long-memory tasks, which we demonstrate across a large variety of tasks. We are excited to see that “idea seems to be quite interesting” and “the presentation of the paper is clear and sound” (R1), with “high potential impact” (R2), and the mechanism is “is elegant and seems novel within Transformer context” (R4). We make two general points:

**(1) Baselines**- Reviewers want to see additional baselines. We have added additional baselines: Adaptive-Span and Compressive Transformer for several tasks in the paper. Our proposed Expire-Span method outperforms both baselines. We’d also like to clarify that our main baseline shown before these additions is actually Transformer-XL. We have updated the paper to make this clear.
We also provide a summary diagram to clarify our contribution. A large number of Transformer variant models have been proposed recently - many of these directly adjust the attention mechanism, which is actually completely orthogonal to our work - our work adjusts not the processing of the current block, but what to remember from previously processed blocks. Expire-Span can be added to any mechanism that adjusts attention, because Expire-Span operates on models that process blockwise and cache the past (hence Transformer-XL being our main baseline).

Models that are baselines are models such as Transformer-XL and Compressive Transformer, as these modify the block-wise processing of long sequences. For example, Compressive Transformer compresses all previous tokens the same amount, while Expire-Span learns what to forget and what to keep salient. Adaptive-Span can adjust, based on the data, the amount of attention context size required, but Expire-Span can operate on drastically longer timescale and has stronger performance even at similar memory sizes.

Please click this link to see our summary figure: https://imgur.com/a/QnRetg8 (hosted on imgur for anonymization)

Finally, we’d like to emphasize that Expire-Span provides the ability for Transformers to scale to very large attention spans. It is unclear if tasks such as Language Modeling actually benefit from this ability --- e.g. do you really need to know a word mentioned 64K words ago? We believe nevertheless that demonstrating this ability is important, and show on various constructed tasks that are designed to require this size of memory. Various other tasks can be proposed in the future that would require such memory size, for example in video processing.

**(2) Memory and Efficiency of Expire-Span** -  Reviewers would like more explicit efficiency comparisons. Below, we provide the peak GPU memory usage and Training time per Batch for Expire-Span compared to various baselines (TransformerXL, Adaptive-Span, and Compressive Transformer). We also provide the comparative results on the evaluation sets. Overall, we demonstrate across various tasks that Expire-Span is more efficient in terms of GPU memory usage and per-batch training time (per suggestion of R2 for these metrics), while achieving better performance. We include results on real tasks, enwiki8 and PG-19, per the suggestion of R1.

**Collision Task**

|Method| Max Span  | Test Error  | Peak GPU Memory  | Train Time/Batch |
|---|---|---|---|---|
| Adaptive-Span  | 16k  | 59.8% | 17.2GB  | 365ms |
| Compressive Transformer  | 8.5k  | 63.8%  | 11.6GB  | 327ms |
| Expire-Span | 16k | 52.2% | 11.9GB | 130ms |

We added a strong Adaptive-Span baseline for the collision task to compare against our method. Adaptive-Span’s performance is slightly worse than our method, but its efficiency is much worse: 44% more memory usage and 2.8x slower to train on a single batch (measure on the final checkpoint). This result is not surprising because the adaptive-span model needs to process all frames after the collision, while the expire-span can forget them and only process the collision frame itself.

We also included a Compressive Transformer baseline with a maximum span of 8.5k (512 normal memories and 2k memories compressed by 4x). This model performs worse than our model, and increasing its span to 16k does not help. It uses the same GPU memory as our 16k model, but runs 2.5x slower, despite having a smaller span. All models are trained in the same setting (32 V100 GPUs).

**Instruction Task**

|Method| Max Span  | Test Accuracy  | Peak GPU Memory  | Train Time/Batch |
|---|---|---|---|---|
| Adaptive-Span  | 1.5k  | 68% | 14.2GB  | 240ms |
| Compressive Transformer  |1.5k  | 71%  | 9.8GB  | 210ms |
| Expire-Span | 1.5k | 74% | 8.4GB | 90ms |

We show that Adaptive-Span has worse performance and worse efficiency. Compressive Transformer uses only marginally more GPU memory, and the performance is not that far behind Expire-Span,  but the training time per batch is much slower --- over double that of Expire-Span. All models are trained in the same setting.

---

> ### Author Response · Authors · 2020-11-23
> **General Response, continued**
>
> continued response on the efficiency of Expire-Span, from above
>
> **Character-level Language Modeling on enwiki8**
>
> |Method| Max Span  | Dev BPC  | Peak GPU Memory  | Train Time/Batch |
> |---|---|---|---|---|
> | Transformer-XL | 2k  | 1.058 | 25.6GB  | 658ms |
> | Compressive Transformer | 4k  | 1.015 | 21.3GB  | 805ms |
> | Adaptive-Span  | 8k  | 1.036  | 20.3GB  | 483ms |
> | Expire-Span | 16k | 1.034 | 14.6GB | 408ms |
>
> We added a comparison of the actual efficiency against three baselines: Transformer-XL, Adaptive-Span and Compressive Transformer. All models are trained on 32 GPUs. Among them, our method is the most efficient in terms for speed and memory usage, and obtains the best performance. The Transformer-XL has a fixed memory that can’t adapt to data, so it quickly becomes slow and memory hungry even at a context size of 2k.  Adaptive-Span can adapt to data to adjust its memory size, so it does much better than the Transformer-XL. But, its memory size is fixed after training, so it lacks the flexibility of Expire-Span where memory size depends on the context. This disadvantage hurts it’s efficiency as shown in the table.
>
> We implemented a mean-pooling version of the Compressive Transformer. It has a worse performance and uses more GPU memory and is slower to train per batch when the memory size is 4k. Increasing the memory size 8k caused out-of-memory.
>
> **Character level language modeling on PG19**
>
> |Method| Max Span  | Dev BPC  | Peak GPU Memory  | Train Time/Batch |
> |---|---|---|---|---|
> | Adaptive-Span  | 16k  | 1.12  | 13.5GB  | 515ms |
> | Expire-Span | 16k | 1.12 | 12.9GB | 585ms |
>
> We added an Adaptive-Span baseline for PG19 to compare to our Expire-Span model. We found that Adaptive-Span has similar performance to Expire-Span, most likely because the Expire-Span span size is too large. We are investigating increasing the Expire-Span loss to reduce the span size. For efficiency, Expire-Span and Adaptive-Span are similar, with Expire-Span using less GPU memory. We weren’t able to train a Compressive Transformer in the given rebuttal period.

---

### Decision · Program_Chairs · 2021-01-07
**Final Decision**

**Decision:**

Reject

**Comment:**

The paper studies the problem of identifying what information to forget in attention mechanisms, with the goal of enabling attention mechanisms to deal with longer contexts. This is a simple yet intuitive extension:  self-attention is augmented with an expiration value  prediction. Experiments were carried out on NLP and RL tasks.
Overall, the paper has novelty in the proposed idea, however, there are concerns about the strength of the experiments; that the experiments fall short.